# Increased activity of procoagulant factors in patients with small cell lung cancer

**Shona Pedersen** [1,2]☯*, **Anne Flou Kristensen** [1,2]☯, **Ursula Falkmer** [2,3], **Gunna Christiansen** [4,5], **Søren Risom Kristensen** [1,2]

1 Department of Clinical Biochemistry, Aalborg University Hospital, Aalborg, Denmark, 2 Department of Clinical Medicine, Aalborg University, Aalborg, Denmark, 3 Department of Oncology, Aalborg University Hospital, Aalborg, Denmark, 4 Department of Biomedicine, Aarhus University, Aarhus, Denmark, 5 Department of Health Science and Technology, Aalborg University, Aalborg, Denmark

☯ These authors contributed equally to this work.
* shp@rn.dk

**Data Availability Statement:** All relevant data are within the paper and its Supporting information files.

**Funding:** This work was funded by grants from the Danish Research Council for Independent

## Abstract

Small cell lung cancer (SCLC) patients have augmented risk of developing venous thrombo-embolism, but the mechanisms triggering this burden on the coagulation system remain to be understood. Recently, cell-derived microparticles carrying procoagulant phospholipids (PPL) and tissue factor (TF) in their membrane have attracted attention as possible contributors to the thrombogenic processes in cancers. The aims of this study were to assess the coagulation activity of platelet-poor plasma from 38 SCLC patients and to provide a detailed procoagulant profiling of small and large extracellular vesicles (EVs) isolated from these patients at the time of diagnosis, during and after treatment compared to 20 healthy controls. Hypercoagulability testing was performed by thrombin generation (TG), procoagulant phospholipid (PPL), TF activity, Protein C, FVIII activity and cell-free deoxyribonucleic acid (cfDNA), a surrogate measure for neutrophil extracellular traps (NETs). Our results revealed a coagulation activity that is significantly increased in the plasma of SCLC patients when compared to age-related healthy controls, but no substantial changes in coagulation activity during treatment and at follow-up. Although EVs in the patients revealed an increased PPL and TF activity compared with the controls, the TG profiles of EVs added to a standard plasma were similar for patients and controls. Finally, we found no differences in the coagulation profile of patients who developed VTE to those who did not, i.e. the tests could not predict VTE. In conclusion, we found that SCLC patients display an overall increased coagulation activity at time of diagnosis and during the disease, which may contribute to their higher risk of VTE.

## Introduction

The risk of venous thromboembolism (VTE) is increased in cancer patients but the underlying mechanisms are not well-known. The risk differs in different types of cancer malignancies, and those in the lung, brain, pancreas and gastrointestinal tract are considered to be associated

Research, 4183-00268), https://ufm.dk/ (S.R.K.);
and the Obel Family Foundation, 26145, http://
www.europeanfunding-guide.eu/scholarship/7862-
obel-family-foundation (S.R.K.). The funders had
no role in study design, data collection and
analysis, decision to publish, or preparation of the
manuscript.

**Competing interests:** The authors have declared
that no competing interests exist.

**Abbreviations:** 100K pel, Pellet obtained at
100,000 x $g$ for 30 min; 20K pel, Pellet obtained at
20,000 x $g$ for 30 min; ALAT, alanine amino
transferase; CAT, Calibrated automated
thrombography; cfDNA, Cell free DNA; DPBS,
Phosphate-Buffered Saline; ED, Extensive disease;
ETP, Endogenous Thrombin Potential; EV,
Extracellular vesicles; FVII, Coagulation factor VII;
FVIII, Coagulation factor VIII; IEM, Immunoelectron
microscopy; LD, Limited disease; NETs, Neutrophil
extracellular traps; Peak, Peak height; PPL,
Procoagulant phospholipids; PPP, Platelet-poor
plasma; SCLC, Small cell lung cancer; SPP,
Standard pool plasma; TEM, Transmission electron
microscopy; TF, Tissue factor; TG, Thrombin
generation; ttPeak, Time to peak; VTE, Venous
thromboembolism.

with the highest incidence of VTE [1–4]. Among patients with lung cancer, the incidence rate
has been reported to be approx. 50 pr. 1000 person years [5], and patients with small cell lung
cancer (SCLC) have been found to have an incidence rate of 31.7 pr. 1000 person years [6].
Chemotherapy and especially platinum-based regimens, which is the preferred treatment strat-
egy for patients with SCLC, may further increase the risk of thrombotic events [4].

The coagulation system consists of a complex interplay of many factors, and several procoa-
gulant factors may be involved in this process [7]. Tissue factor (TF), the main initiator of
coagulation, has been found to be increased in plasma of some cancer patients compared to
healthy controls [8–10]. TF is an integral membrane bound protein which has been proposed
to be expressed by cancer cells where it is related to the metastatic potential of these cells but
may also participate in the cancer-associated hypercoagulability [11]. TF may be present in
plasma, carried in the membranes of extracellular vesicles (EV) possibly released from the can-
cer cells, and this EV-associated TF activity may play a major role in cancer-related thrombo-
genicity [7, 12–14]. EV-associated TF activity in cancer patients has also been correlated to
cancer prognosis [8, 15, 16]. In addition, procoagulant phospholipids (PPL), i.e. anionic phos-
pholipids, mainly phosphatidylserine, and a high level of coagulation factor VIII (FVIII),
which is an acute-phase protein, have been associated with an increased risk of thrombosis
[17, 18] also including cancer patients [7, 19–22]. PPL is also present on the surface of EVs in
plasma. Thus, an increased level of EVs containing TF and PPL may play a crucial role for the
risk of VTEs in cancer patients [14, 23–25].

Neutrophil extracellular traps (NETs), have been linked to the formation of VTEs, and may
also play a role in cancer-associated thrombosis [26–28]. During NETosis a network of chro-
matin is extruded through the membranes of activated neutrophils, which may trap and acti-
vate platelets and coagulation factors, and may thus initiate thrombosis [18, 29, 30]. Although
there is no single marker to determine NETs in plasma, surrogate markers for NETs include
plasma levels of cfDNA, citrullinated histone H3 and myeloperoxidase [18, 31]. In addition,
cfDNA was described to carry a procoagulant activity on its own and therefore might give
insight about NET-associated procoagulant activity in patient plasma samples [18, 32, 33].
Extracellular vesicles (EVs) are diverse membranous vesicles that are actively released by many
types of cells including tumor cells and, have been associated with several pro-tumoral pro-
cesses [30, 34–36]. Two main types of EVs have been identified in the biological fluids of can-
cer patients: exosomes (small EVs) and microvesicles (MVs-large EVs) [37]. Exosomes, i.e.
small EVs, produced by cancer cells have been shown to stimulate NETosis in a mouse model
[30]. Protein C is an important anticoagulant factor inactivating the activated FV and FVIII
(FVa and FVIIIa, respectively). A low protein C activity was associated with increased mortal-
ity in different malignancies, e.g. in non-metastasizing lung cancer patients [38] and low levels
of protein C has been found to be linked to cancer-related VTE [21].

The aim of the present study was to investigate the coagulation profile of SCLC patients at
the time of diagnosis, during and after treatment. Coagulation activity was assessed by throm-
bin generation (TG), procoagulant phospholipid (PPL) and TF activity, Protein C, FVIII activ-
ity and cfDNA. Furthermore, we investigated the effect on coagulation of plasma-derived EVs
isolated from SCLC patients compared to that of healthy controls. We hypothesize that these
patients have an overall increased coagulation activity and that this may be, at least partially,
mediated by the presence of procoagulant EVs.

## Materials and methods

### Study design and patient demographics

Thirty-eight patients newly diagnosed with SCLC were enrolled in this study after obtaining informed consent. The study was conducted in agreement with the Declaration of Helsinki and approved by the Regional ethics committee for Northern Jutland (N-20140055). Inclusion criteria for enrolment were: age over 18 years, patients should be eligible to receive treatment with chemotherapy consisting of platinum and a topoisomerase inhibitor, platelet count >100 x109/L and normal levels of INR and APTT. In all cases, histological and/or cytological confirmation SCLC was required. Exclusion criteria were: prior systemic chemotherapy for lung cancer, concomitant anticoagulation treatment (platelet inhibitors, ASA and clopidogrel were allowed), active or at high risk of overt bleeding of clinical importance, severe coagulopathy such as haemophilia, severe liver dysfunction with impaired coagulation, acute peptic ulcer, occurrence of intracranial haemorrhage 3 months prior to start of the study, or surgery in the central nervous system. Pregnancy and/or breast-feeding and women who did not use oral contraceptives were excluded. Treatment with any other investigational agents, or participation in other clinical trials also lead to exclusion. Patients were staged according to limited disease (LD) or extensive disease (ED), equivalent to disease limited to one hemithorax and disease beyond the ipsilateral hemithorax, respectively [39]. The study group had access to patient files including routine laboratory results and medical history. In addition, a total of 20 healthy age-related controls with a mean age of 63 years (range 56–67, 11 male and 9 female) were included in the study at the Blood Donor Center at Aalborg University Hospital. In Denmark blood donors are healthy volunteers without any apparent illness and without biochemical abnormalities.

### Blood sample collection

Patients donated 3 blood samples: one at baseline prior to initiation of treatment cycle 1 (denoted baseline); one prior to initiation of treatment cycle 3 (denoted "during treatment"); and the last one at a follow-up visit scheduled 3 weeks or 2 months after completion of the sixth cycle (denoted follow-up). Blood samples were drawn in the antecubital vein using a vacutainer blood collection device with a 21-gauge butterfly needle. Blood was collected in 6 ml 3.2% (109 mM) sodium citrate tubes (Vacuette® Greiner Bio-One, Austria) and the first few millilitres, i.e. the first collection tube, were discarded. Samples were centrifuged twice at 2500 x g for 15 min to obtain a platelet-poor plasma (PPP). After the first centrifugation plasma was collected to 1 cm above the buffy coat, and after the second centrifugation the last 0.5 ml was left in the tube. Aliquots of 1 ml of PPP were immediately frozen and stored at -80 ˚C until further analysis. Standard plasma (SP) for EV analyses was obtained from one healthy donor with centrifugation performed as described above.

### Isolation of extracellular vesicles

Extracellular vesicles (EVs) were isolated from patient 1–30 and all healthy controls using differential ultracentrifugation in the following manner: 1 ml PPP was centrifuged at 20,000 x g for 30 min. (pellet denoted 20K pel), and the supernatant was then re-centrifuged at 100,000 x g for 1 h (pellet denoted 100K pel). Both pellet types were subjected to a washing step in 1 mL Dulbecco's Phosphate-Buffered Saline (DPBS), and subsequently centrifuged as the initial centrifugation. Pellets were either resuspended in 200 μl SPP or DPBS, or in 180 μl HBSA buffer (137 mM NaCl, 5.38 mM KCl, 5.55 mM D-glucose, 10 mM HEPES, 0.1% bovine serum albumin; pH 7.4) dependent on analyses.

## Thrombin generation analysis

TG determinations on each plasma sample as well as SP containing isolated EVs from both patients and healthy controls were determined using the CAT assay developed by Hemker et al [40]. In short, 80 μL plasma was dispersed into a well containing either 20 μL trigger or calibrator solution, incubated 10 minutes at 37˚C, and subsequently coagulation was initiated by the addition of 20 μl FluCa buffer containing a fluorescence substrate and $CaCl_2$ (all reagents from Thrombinoscope BV, Netherlands). For the analysis of plasma, a commercially available trigger denoted PPPlow, containing 1 pM TF and 4 μM phospholipids (Thrombinoscope BV, Netherlands) was used. Analysis of EVs was performed using the trigger reagent PRP, containing 1 pM TF only (Thrombinoscope BV, Netherlands). Fluorescence intensity was recorded for a total of 50 minutes with a 390/460 nm excitation/emission filter set and thrombograms were generated using Thrombinoscope software version 5.0 (Thrombinoscope BV, Netherlands). TG parameters were assessed in the statistical analysis i.e. lag time, Endogenous Thrombin Potential (ETP), peak height (peak), and time to peak (ttPeak).

## STA Procoag-PPL assay

To determine the activity of PPL both in plasma samples and in SPP containing isolated vesicles, the commercially available STA Procoag-PPL test (Diagnostica Stago, Asnieres, France) was applied. To determine the activity of procoagulant phospholipids (i.e. Procoag-PPL), an activated factor X (FX)-based clotting method is used. The assay is initiated by the addition of 25 μL PPP to a cuvette containing 25 μL human phospholipid depleted plasma. The sample was then incubated for 2 min. at 37˚C, prior to the addition of pre-warmed XACT reagent containing FXa and $Ca^{2+}$ (Diagnostica Stago, Asnieres, France). Clotting time is determined by the motion of a spherical steal ball; a short clotting time indicates a high activity of procoagulant phospholipids.

## Activity of extracellular vesicle-associated tissue factor

The activity of EV-associated TF in both pellet types i.e. 20K pel and 100K pel, was determined using a method described by Hisada and Mackman [41]. The analysis was as follows: Anti-human HTF-1 antibody (4μg/ml, BD Pharmingen, CA, USA) or mouse control IgG antibody (4μg/ml, BD Pharmingen, CA, USA) was added to 40 μl isolated EVs resuspended in HBSA buffer in a 96 well plate and incubated at room temperature for approx. 15 min. After incubation, 50 μL HBSA containing $CaCl_2$ (10 mM) and coagulation factors VIIa (10 nM) and X (300 nM) were added to each well. A standard curve of a recombinant human TF (0.63–30 pg/ml Innovin, Siemens, Germany) was applied to the plate, which was incubated for 2 hours at 37˚C. FXa generation was terminated by 25 μL HBSA containing EDTA (25 nM), and 25 μL of Pefachrome FXa (FXa8595, Pentapharm, Switzerland) substrate was added to all wells. The plate was then incubated for 15 min at 37˚C and absorbance was measured at 405 nm using a FLUOstar Optima microplate reader (BMG labtech, Germany). The EV-associated TF-dependent FXa generation was calculated by subtracting the FXa generation in the HTF-1 wells from the FXa generation in the control IgG wells. According to Hisada et al [9] an EV-related TF activity of <1.0 pg/ml is considered to be weak, while >1.0 pg/ml represents a moderate to strong activity.

## Plasma cell free DNA

Formation of NETs was determined using plasma cfDNA as a surrogate marker. Plasma was diluted 10-fold and allocated into a 96 well plate, where 100 μl of either Sytox Green (1: 1250,

Invitrogen, CA, USA), a fluorescent DNA dye, or DPBS was applied to the plasma samples. Samples where only DPBS was added, were blanks to correct for fluorogenic background noise originating from the samples. Calf thymus DNA (0.0–125 ng/ml, Invitrogen, CA, USA) was used as a standard curve to determine the concentration of DNA in each sample. After five minutes of incubation at 27˚C, fluorescence intensity was measured using a 485/520 nm excitation/emission filter set in a FLUOstar Optima microplate reader (BMG Lagtech, Germany).

## Protein C and coagulation factor VIII activity

Protein C and FVIIIa were measured on an ACL TOP 500 CTS coagulation analyzer (Instrumentation Laboratory, Ma, USA) with dedicated reagents (Instrumentation Laboratory, Germany) using Protein C deficient plasma or FVIII deficient plasma, respectively.

## Determination of vesicles in the isolates

Determination and confirmation of EVs was performed in accordance with guidelines recommended by International Society of Extracellular Vesicles (ISEV) [42, 43]. To characterize EVs, Nanoparticle Tracking Analysis was applied to determine the size distribution and vesicle concentration in the suspension, western blotting analysis was used to validate the isolated vesicles for the common EV marker CD9, and transmission electron microscopy (TEM) and immunoelectron microscopy (IEM) were applied for the structural characterization of the isolated EVs with immune-gold labelling against CD9.

## Nanoparticle tracking analysis

A LM10-HS Nanoparticle Tracking Analysis system (Malvern Instruments Ltd, UK) equipped with a 405 nm laser and a Luca-DL EMCCD camera (Andor Technology, UK) was used to determine the concentration and size of particles in the pellets. Samples were diluted in PBS to obtain an average of 17–80 particles per frame and a total of 5×30 sec videos were captured for each sample. For sample analysis camera level 11, detection threshold 2, and blur 9×9 were employed. Analysis was performed using Nanoparticle Tracking Analysis software version 3.0 (Malvern Instruments Ltd, UK). Standard 0.1 μm silica beads (Polysciences, Germany) were used to validate settings.

## Western blotting

Western blotting was performed on a pool of isolated vesicles from controls or patients i.e. for each sample 12 μl pooled vesicles resuspended in PBS was used. Proteins were separated in MiniProtean TGX 4–15% gels (Bio-Rad, Denmark) using Laemmli sample buffer (1:1.6, Bio-Rad, Denmark) for 50 min. at 150 Volt under non-reducing conditions. Proteins were transferred to an Amersham Hybond 0.45 PVDF Blotting membrane (GE Healthcare, Denmark) for 60 minutes at 100 Volt and blocked in a 5% (w/v) skim milk Tris-Glycine buffer. Primary antibody was mouse monoclonal anti-human CD9 (clone M-L13, BD Pharmingen, CA, USA), which was diluted 1:1000 in 5% skim milk Tris-Glycine buffer and incubated overnight at 4 ˚C with the membrane. Secondary labelling was performed with a horseradish peroxidase-conjugated anti-mouse antibody (1:30000, Dako, Glostrup, Denmark) for 2 hours at room temperature. ECL Lumi-Light Western Blotting substrate detection reagent (Roche, Schweiz) was used for development and detection was performed on a PXi 4 system (Syngene, UK) with the GeneSys software 1.5.4.0 (Syngene, UK). To determine a relative change in band intensity between samples ImageJ 1.50r software (NIH, Bethesda, USA) was used.

## Transmission electron microscopy and Immunoelectron microscopy of extracellular vesicles

EVs were phenotypically and structurally characterized by Transmission Electron Microscopy (TEM) with immuno-gold labelling against CD9, as previously described by Nielsen et al [44]. Briefly, 5 μl of EV isolate was mounted on a grid (SPI Supplies, PA, USA) and stained with one drop of 1% (w/v) phosphotungstic acid (pH 7.0, Ted Pella, Caspilor AB, Sweden), and subsequently blotted dry on filter paper. To visualize the presence of EV-specific marker CD9 on the surface of vesicles, IEM was performed on the isolated vesicles. The pelleted vesicles were positioned on a grid as described above and then blocked in ovalbumin. Subsequently, the grid was incubated with primary anti-CD9 antibody (1:50, BD Pharmingen, CA, USA), followed by incubation with secondary goat anti-mouse antibody conjugated with 10 nm colloidal gold (1:25, British BioCell, UK). The grids were stained with 1% (w/v) phosphotungstic acid at pH 7.0 and blotted dry. Images were obtained with a transmission electron microscopy (JEM 1010, Germany) operated at 60 keV coupled to an electron-sensitive CCD camera (KeenView, Olympus, PA, USA). Lastly, a grid-size replica (2,160 lines/mm) was imported to ImageJ 1.50i software, which enables a correct determination of the size of EVs.

## Statistics

All data was tested for normality using Shapiro-Wilk normality test prior to statistical analysis. If data assumed a normal distribution a parametric t-test was applied to test for statistical difference between two variables, while a repeated measures ANOVA was used to test for differences between repeated measures i.e. measurements performed over time. The non-parametric Kruskal-Wallis was used if data did not assume a normal distribution and multiple groups were to be compared, while Mann-Whitney U test was applied to non-parametric data if only two groups were to be compared i.e. measurements performed during the disease or measurements between two groups of patients. Correlations between markers was performed using Pearsons correlation test if data assumed a normal distribution, and if not, Spearman correlation was used. Correlation between ordinal and metric data was performed using Eta2 test i.e. the correlation between performance status and all other valuables, where a high degree of correlation was assigned to an Eta2 value above 0.10. For all other analyses, statistically significant difference was assigned to $p < 0.05$. The Eta2 test was performed using IMB SPSS Statistics 23 (SPSS, Chicago, IL, USA), and Graph Pad Prism 6 (GraphPad Software, La Jolla, CA, USA) was used for all other statistical analyses.

## Results

### Patient characteristics

A total of 38 newly diagnosed patients with small cell lung cancer were included, basic characteristics of the patients at inclusion are described in Table 1. From all 38 patients, a blood sample was collected at baseline; during treatment blood samples were collected from 33 patients (3 patients died and 2 withdrew their informed consent); and at the follow-up visit blood samples were collected from 28 patients (3 died, 1 did not wish to donate another sample and 1 relapsed before the follow-up visit). At baseline, all patients had haemoglobin and blood cell counts (leukocytes, and platelets) within normal reference ranges and all patients had normal sodium, potassium, calcium, creatinine, and albumin (S1 Table). One patient had signs of liver disease (increased levels of alanine amino transferase (ALAT), alkaline phosphatase, and lactate dehydrogenase). The patient group included 9 with LD and 29 with ED. In the group with ED four patients had no metastasis, (a large tumor burden evolving beyond the diaphragm

**Table 1. Characteristics of small cell lung cancer patients at diagnosis.**

| Characteristics | SCLC cohort (N = 38) | Healthy controls (N = 20) |
|---|---|---|
| **Age at baseline–years** | | |
| Mean (SD)* | 65 (8.5) | 63 (2.8) |
| Range | 42–80 | 57–67 |
| **Gender–no. (%)** | | |
| Female | 22 (58) | 11 (55) |
| Male | 16 (42) | 9 (45) |
| **Performance status–no. (%)** | | |
| 0–1 | 22 (58) | |
| 2–3 | 16 (42) | |
| **Disease stage–no. (%)** | | |
| Limited disease | 9 (24) | |
| Extended disease | 29 (76) | |
| **T-stage–no. (%)** | | |
| T1 | 5 (13) | |
| T2 | 4 (10) | |
| T3 | 3 (8) | |
| T4 | 17 (45) | |
| TX | 9 (24) | |
| **N-stage–no. (%)** | | |
| N0 | 3 (8) | |
| N1 | 3 (8) | |
| N2 | 3 (8) | |
| N3 | 19 (50) | |
| NX | 10 (26) | |
| **M-stage–no. (%)** | | |
| M0 | 12 (32) | |
| M1 | 26 (68) | |
| **Chemotherapy cycles given–no. (%)** | | |
| < 4 cycles | 3 (8) | |
| ≥ 4 cycles | 35 (92) | |
| **Patients receiving radiotherapy–no. (%)** | | |
| Prophylactic cranial radiation | 19 (50) | |
| **Days Baseline-Follow-Up** | | |
| Median | 145 | |
| Range | 107–259 | |

Two patients withdrew their informed consent during the study period. However, these patients consented to the inclusion of their data in the study.

*Mean and standard deviation.

TNM staging = **T**umor, lymph **N**ode, and **M**etastasis.

places them in the ED group), however, in our study we did not separate patients according to metastases as it will not impact the result outcome. From the TNM (**T**umor, lymph **N**ode, and **M**etastasis) staging we found approximately 45% of the patients to be classified as stage IV [45]. Age- and sex-related healthy blood donors from the blood bank at Aalborg University Hospital were used for comparison.

## Coagulation activity of PPP from small cell lung cancer patients compared to healthy controls

Baseline coagulation data based on analyses of PPP from all SCLC patients and healthy controls are presented in Table 2. The patients had significantly elevated TG with a higher ETP and peak, even though lagtime was slightly longer in the patients. PPL clotting time was considerably shorter indicative of a higher PPL activity in the patients when compared to healthy controls. The concentration of cfDNA was significantly elevated among patients. Factor VIII and protein C activities in patients were increased by 45% and 19%, respectively, compared to controls. In the patients, plasma PPL activity assay correlated significantly with plasma TG parameters: Peak (rho = 0.27, P = 0.015) and ETP (rho = 0.40, P = 0.0003). FVIII activity correlated with peak (rho = 0.37, P = 0.001), whereas the activity of protein C correlated significantly with lag time (rho = 0.21, P = 0.05) and inversely with the activity of FVIII (tau = 0.26, P = 0.041).

## The influence of disease stage and performance status

Table 2 also summarizes data comparing patients with LD and ED, i.e. investigation of the effect of disease stage. There were no differences in TG or PPL activity assay, but patients with ED had a significantly higher concentration of cfDNA compared to patients with LD (P = 0.039) at baseline. A patient performance status of 2–3 corresponding to patients who were more ill, had a high correlation to an increased TG in plasma for all CAT parameters (Eta2 > 0.11) except ETP.

## Coagulation activity in plasma during the disease

There were no substantial changes between the groups from baseline to the samples during treatment and to follow-up (Table 2). However, comparison of the differences of individual patients between baseline samples and "during treatment" and between baseline and follow-up samples, respectively, indicated a shorter lagtime in patients with ED and a trend towards a reduced TG and PPL activity in plasma among patients with LD during treatment and at follow-up (Table 3). TF activity in the 20K pel of the patients with SCLC was 9-fold increased when compared to the pellet extracted from healthy persons (borderline significant (p = 0.51), Table 2) at baseline, but TF was further increased during the disease, most pronounced among patients with LD (Table 3). FVIII activity also showed a propensity to increase in the LD group, while cfDNA tended to decrease in the ED group (Table 3).

## Extracellular vesicles (EVs) isolated using differential centrifugation and their procoagulant effect

Western blot analysis and IEM showed CD9 positive vesicles in both pellet types (Fig 1A–1D). TEM showed the presence of round vesicle-like structures, some with a clear phospholipid bilayer indicating EVs. The original IEM and WB images are displayed in the S1A–S1C and S2 Figs, respectively).

The effect of the pellets on the coagulation activity was investigated. SP in the absence of EVs, had a rather long lagtime and a small peak height, but the addition of the pellets to the SP resulted in shortened lagtime and increased peak height. There was a large variation of this effect between subjects but the mean TG of isolated EVs (30 out of 38 patients) did not differ much from the effect of EVs isolated from healthy age-related controls (Fig 2A), but a trend towards a higher activity of PPL in EVs isolated from patients was found, although not significant (p-value of 0.068 for the 20K pel and 0.059 for the 100K pel) (Fig 2B). EV associated TF

**Table 2. Comparison between healthy controls and SCLC patients at baseline.**

|  | Healthy controls (N = 20) | All SCLC patient (N = 38) |
|---|---|---|
| Lag time (min) | 7.1 ± 1.5 | 8.2 ± 1.8* |
| ETP (nM*min) | 1060.3 ± 270.4 | 1727.2 ± 522.8**** |
| Peak (nM) | 118.7 ± 54.4 | 260.9 ± 113.6**** |
| TF (pg/ml) 20K pel | 0.05 ± 0.14 | 0.44 ± 0.85 |
| PPL (sec) | 61.5 ± 9.7 | 48.7 ± 10.7**** |
| FVIIIa (U/ml) | 0.84 ± 0.53 | 1.54 ± 0.60** |
| Protein C (U/ml) | 1.21 ± 0.25 | 1.50 ± 0.34* |
| cfDNA (ng/ml) | 226.5 ± 54.8 | 376.0 ± 144.0**** |
| | SCLC patients at baseline | |
| | LD (n = 9) | ED (n = 29) |
| Lag time (min) | 8.1 ± 1,7 | 8.2 ± 1.9 |
| ETP (nM*min) | 1809.2 ± 681.2 | 1700.8 ± 473.2 |
| Peak (nM) | 248.5 ± 130.2 | 255.5 ± 110.0 |
| TF (pg/ml) 20K pel | 0.13 ± 0.20 | 0.52 ± 0.93 |
| PPL (sec) | 46.3 ± 11.5 | 49.5 ± 10.5 |
| FVIIIa (U/ml) | 1.40 ± 0.56 | 1.58 ± 0.61 |
| Protein C (U/ml) | 1.61 ± 0.31 | 1.47 ± 0.35 |
| cfDNA (ng/ml) | 315.4 ± 118.4 | 398.8 ± 148.3* |
| | SCLC patients during treatment | |
| | LD (n = 9) | ED (n = 24) |
| Lag time (min) | 8.0 ± 4.4 | 6.5 ± 1.3 |
| ETP (nM*min) | 1747.0 ± 854.6 | 1694.1 ± 479.5 |
| Peak (nM) | 236.1 ± 89.3 | 223.2 ± 69.7 |
| TF (pg/ml) 20K pel | 0.50 ± 1.01 | 0.94 ± 1.80 |
| PPL (sec) | 49.8 ± 13.2 | 46.75 ± 10.1 |
| FVIIIa (U/ml) | 1.72 ± 0.39 | 1.45 ± 0.44 |
| Protein C (U/ml) | 1.59 ± 0.41 | 1.49 ± 0.33 |
| cfDNA (ng/ml) | 281.8 ± 51.1 | 308.7 ± 93.8 |
| | SCLC patients at follow-up | |
| | LD (n = 7) | ED (n = 21) |
| Lag time (min) | 7.1 ± 2,9 | 6.8 ± 1.4 |
| ETP (nM*min) | 1473.2 ± 390.8 | 1676.6 ± 613.5 |
| Peak (nM) | 192.1 ± 58.7 | 217.3 ± 92.2 |
| TF (pg/ml) 20K pel | 0.91 ± 1.16 | 0.44 ± 0.66 |
| PPL (sec) | 54.5 ± 11.2 | 51.9 ± 15.2 |
| FVIIIa (U/ml) | 1.60 ± 0.36 | 1.42 ± 0.47 |
| Protein C (U/ml) | 1.69 ± 0.47 | 1.41 ± 0.37 |
| cfDNA (ng/ml) | 325.7 ± 119.7 | 274.9 ± 85.0 |

Moreover, patients were further divided into limited (LD) or extended (ED) disease and, followed during treatment and follow-up.

All data are shown as mean ± standard deviations (SD). Student *t*-test or Mann-Whitney U test, depending on the distribution type, was applied to compare the different parameters from the healthy control group to the LD and ED patients, respectively. Significant levels resemble statistical difference from the healthy control group.

* $P < 0.05$;

**$P < 0.01$;

*** $P < 0.001$;

**** $P < 0.0001$.

**Table 3. Percentage deviation from baseline in all the different measured values during treatment.**

| | Limited disease | | Extended disease | |
|---|---|---|---|---|
| | *Ba–Cyc (N = 9)* | *Ba-FU (N = 7)* | *Ba-Cyc (N = 24)* | *Ba-FU (N = 19)* |
| Δ Lag time (min) | -2.6% ±43.1% | -8.5% ±40.4% | -20.4%*** ±18.7% | -13.7% ± 28.0% |
| Δ ETP (nM*min) | -10.7% ±20.9% | -18.6% ±22.1% | 5.8% ±39.1% | 16.6% ±75.2% |
| Δ Peak (nM) | -16.3% ±19.8% | -22.4% ±39.6% | 8.7% ±56.0% | 22.6% ±128.7% |
| Δ PPL (sec) | 15.2% ±52.6% | 31.5% ±68.9% | -4.4% ±21.0% | 6.0% ±32.2% |
| Δ TF activity (pg/ml) 20K pel | -0.34 ±1.09 | -0.69 ±1.28 | -0.38 ±2.25 | -0.03 ±1.79 |
| Δ FVIIIa (U/ml) | 50.3% ±111.5% | 58.1% ±123.0% | 11.6% ±40.5% | 25.7% ±58.7% |
| Δ Protein C (U/ml) | 0.4% ±25.8% | 8.7% ±21.7% | 2.8% ±16.6% | -7.9% ±22.4% |
| Δ cfDNA (pg/ml) | -3.9% ±28.1% | 9.1% ±38.8% | -16.3% ±28.9% | -32.5% ±20.9% |

Patients are divided into limited disease and extended disease. Data is presented as the mean percentage deviation from baseline of all patients in each group ± standard deviation (SD) for all measurements except the tissue factor activity, which is denoted as a difference in activity from baseline to during treatment ± SD. Kruskal-Wallis or independent *t*-test was used to depict the significant changes of the different parameter levels from baseline, noted as

* $P<0.05$;

*** $P<0.001$.

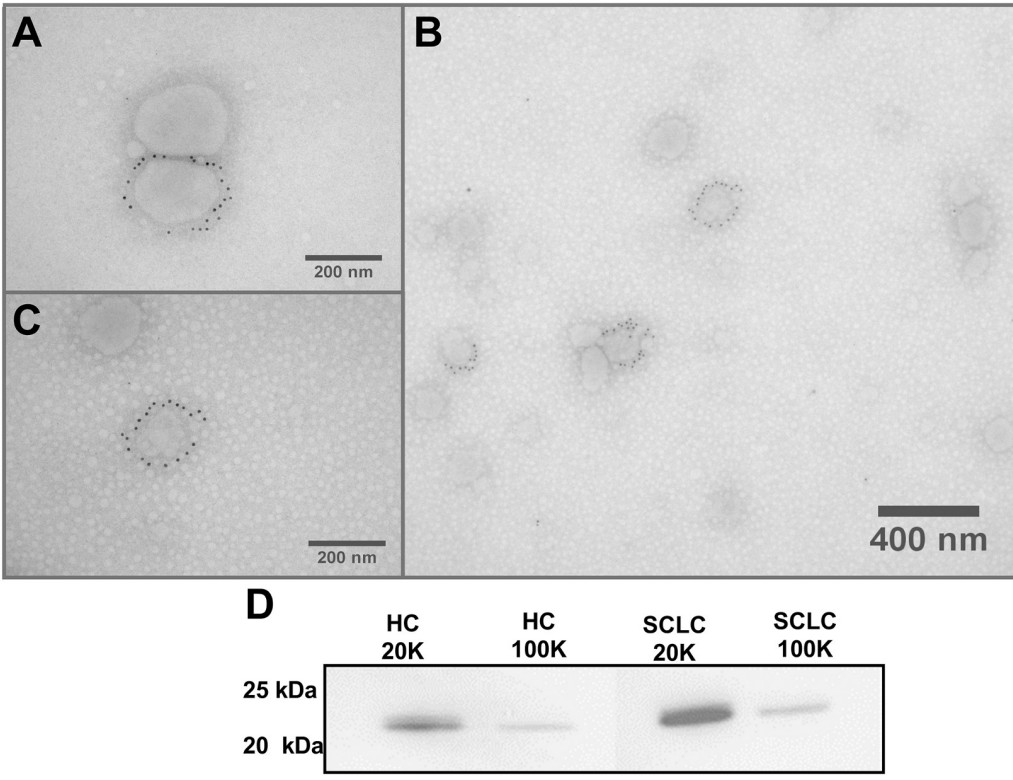

**Fig 1. EV confirmation and validation.** Immunoelectron microscopy (IEM) and Western blot analysis of extracellular vesicle marker CD9 performed on a pool of isolated vesicles from all donors. A) 20K pel CD9 positive vesicles. B) Both CD9 positive and negative vesicles isolated for the 100K pel. C) 100K pel CD9 positive vesicle. D) Western blot analysis against CD9 for the 20K and 100K EV pellets from healthy controls (HC) and small cell lung cancer patients (SCLC).

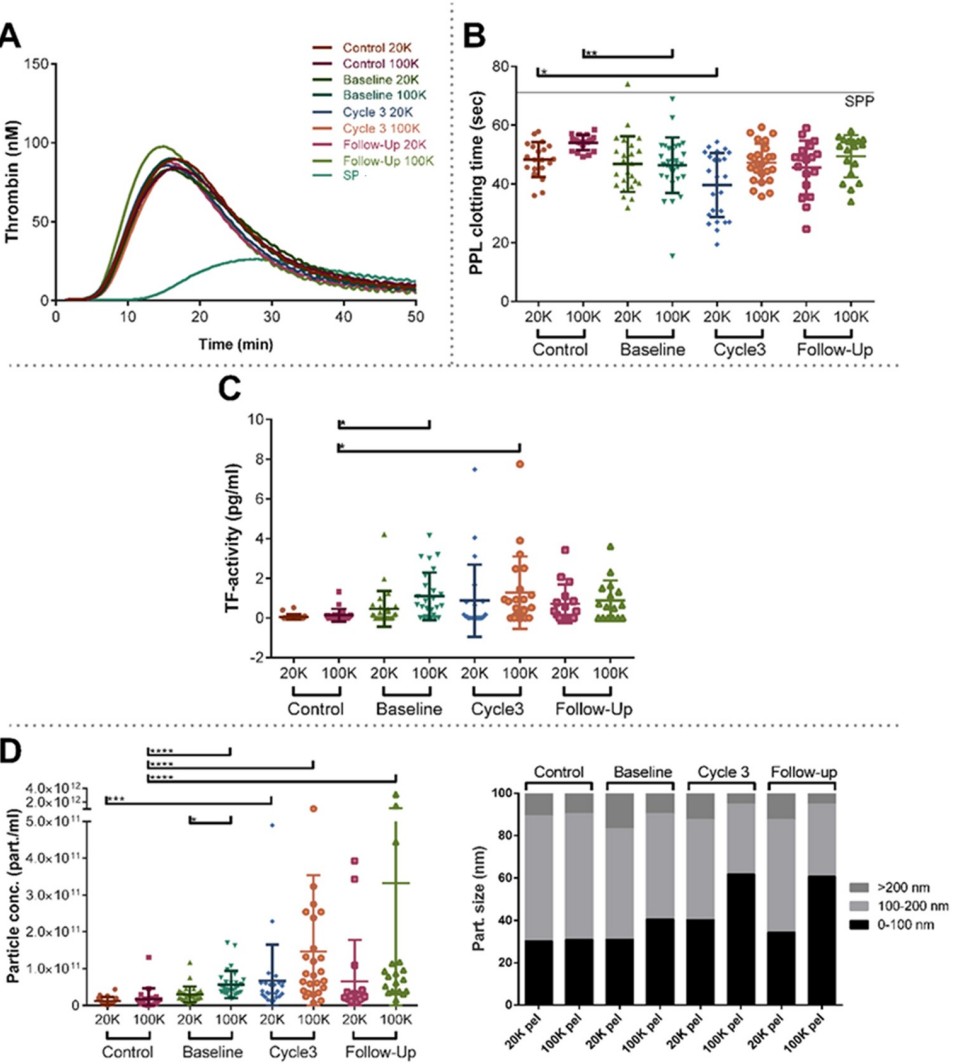

**Fig 2. Procoagulant profiling of EVs isolated from healthy age-related controls and SCLC patients at baseline, during (prior to cycle 3) and after treatment (follow-up).** A) Thrombin generation curves from standard plasma (SP) and SP-containing vesicles isolated from patients and controls. B) EV-associated activity of procoagulant phospholipids (PPL) depicted as a clotting time. In both A) and B) SP represents the coagulation activity of the standard pooled plasma into which the isolated vesicles have been added. C) difference in tissue factor (TF) activity associated with EVs D) concentration and size distribution of isolated vesicles in both pellet types i.e. 20K pel and 100K pel.

activity in both pellet types were increased in patients; with the highest activity in the 100K pel (Fig 2C). Concurrently, in the control group for the 20K sample, none of the donors had a TF activity >1 pg/ml, whereas five percent (i.e. one donor), had a TF activity >1 pg/ml in the 100K sample. More patients had a TF activity >1 pg/ml: at baseline 10% and 33%, during treatment 16% and 32%, and at follow-up 25% and 35% in the 20K pel and 100K pel, respectively. Particle concentrations were higher in patients compared to controls (Fig 2D), and in 20K pel EVs were larger in the patients (Fig 2D). However, more small-sized particles were present in the 100K pel, and the fraction of small particles increased during the treatment (Fig 2D).

## Coagulation activity in plasma associated with extracellular vesicles

The activity of PPL in plasma correlated significantly with PPL activity in both pellet types: for 20K pel rho = 0.50 (p < 0.0001) and for 100K pel rho = 0.50 (p < 0.0001); PPL in plasma also correlated with TG of the 20K pel: rho = -0.41 for peak (p = <0.0001) and rho = 0.45 for lag time (p = 0.001). Moreover, we found a minor but significant correlation between TG measured in plasma and PPL activity in both pellet types: For 20K pel peak vs PPL rho = -0.26 (p = 0.05) and ETP vs PPL rho = -0.29 (p = 0.028); and for 100K pel peak vs PPL rho = -0.34 (p = 0.006) and ETP vs PPL rho: -0.27 (p = 0.03).

## Thrombotic events

During the study period, a total of 4 patients (two males and two females) developed a venous thrombosis, all of them pulmonary emboli. Fig 3 shows the coagulation profile for each of these patients. One patient developed VTE before the sample drawn during treatment (patient 1), two patients after the blood sample during treatment (patients 2 and 3), and the fourth developed VTE after the last treatment cycle. Three of the four patients had ED and had metastases (patients 2, 3, and 4). At baseline, laboratory investigations and characteristics were comparable to the other patients. Patient 1 had a considerably increased FVIII and protein C during treatment and at follow-up, which was after the VTE event. Patient 2 had a marked increase of TG in plasma and a shorter PPL clotting time during treatment, i.e. before the patient developed VTE, and this persisted at follow-up. Unfortunately, TG in plasma from patient 4 could not be measured during treatment because the patient was anticoagulated.

## Discussion

The aim of the study was to investigate the coagulation activity in patients with SCLC at diagnosis and during antineoplastic treatment. The results show that coagulation activity is significantly increased in the patients compared to age-related healthy controls, but coagulation activity did not change substantially during treatment and at follow-up. Although EVs in the patients revealed an increased PPL and TF activity compared with the controls, the TG profiles of EVs added to a standard plasma were similar for patients and controls. Finally, we found no differences between patients who developed a VTE and the other SCLC patients, i.e. the tests could not predict VTE.

In this study, we used thrombin generation as a global test of the coagulation activity. Many factors in plasma contribute to the level: a high level of TF will tend to shorten lagtime; a high level of PPL will tend to increase ETP and Peak; high levels of other coagulation factors will also tend to increase ETP and peak whereas coagulation inhibitors may lower ETP and peak and perhaps increase lagtime. However, using PPPlow reagent containing both TF and PPL will attenuate the effect of these factors present in plasma. A high FVIII will tend to increase TG whereas a high Protein C will tend to reduce it, although the effect is limited in the absence of thrombomodulin. When EVs are isolated and added to SP it will mainly be TF and PPL in the EVs that have effect and using the trigger reagent PRP, which is devoid of PPL, will increase the sensitivity for PPL. We found that plasma from SCLC patients hold a hypercoagulable profile with increased TG and FVIII activity, and a significantly decreased PPL clotting time, indicating a high PPL activity, compared to a group of healthy age-related controls. Furthermore, TF activity was increased in the patients (borderline significant). Interestingly, we also observed the activity of coagulation inhibitor protein C to be higher among the patients (i.e. not decreased as reported by Tafur et al [21]. The activity of protein C correlated with a longer lagtime, whereas the activity of FVIII positively correlated with the peaks. This may imply that the increase in protein C may be a compensatory effect to diminish the

| Patients who developed a VTE | | | | | |
|---|---|---|---|---|---|
| **Baseline** | Pt 1 | Pt 2 | Pt 3 | Pt 4 | Other pt's |
| ETP (nM*min) | 1361.6 | 1816.5 | 1583.3 | 1576.6 | 1744.5±249.0 |
| Peak (nM) | 158.7 | 293.4 | 273.5 | 191.6 | 264.7±118.2 |
| PPL (sec) | 56.7 | 41.9 | 43.7 | 56.8 | 48.5±11.0 |
| TF activity (pg/ml) 20K pel | 0.56 | <0.1 | <0.1 | <0.1 | 0.48±0.93 |
| FVIIIa (U/ml) | 0.50 | 1.03 | 1.74 | 1.24 | 1.60±0.59 |
| Protein C (U/ml) | 1.46 | 0.94 | 1.04 | 1.74 | 1.52±0.34 |
| cfDNA (pg/ml) | 323.1 | 288.5 | 331.5 | 193.4 | 376.0±144.0 |
| | | VTE | | | |
| **During treatment** | | | | | |
| ETP (nM*min) | N/A | 2428.2 | 1511.3 | - | 1687.1±580.6 |
| Peak (nM) | N/A | 271.7 | 256.6 | - | 223.5±75.6 |
| PPL (sec) | 44.7 | 36.3 | 44.2 | - | 48.5±11.4 |
| TF activity (pg/ml) 20K pel | 0.17 | <0.1 | 0.44 | - | 0.91±1.86 |
| FVIIIa (U/ml) | 2.06 | 0.83 | 1.74 | - | 1.53±0.42 |
| Protein C (U/ml) | 2.32 | 1.08 | 1.22 | - | 1.52±0.32 |
| cfDNA (pg/ml) | 257.9 | 232.1 | 231.5 | - | 300.1±82.5 |
| | | VTE | VTE | | |
| **Follow-up** | | | | | |
| ETP (nM*min) | 1547.6 | 2379.5 | - | - | 1571.5±546.6 |
| Peak (nM) | 156.9 | 291.7 | - | - | 207.3±83.1 |
| PPL (sec) | 51.7 | 37.1 | - | - | 53.7±14.1 |
| TF activity (pg/ml) 20K pel | 0.77 | 1.23 | - | - | 0.71±1.01 |
| FVIIIa (U/ml) | 1.95 | 1.70 | - | - | 1.45±0.44 |
| Protein C (U/ml) | 2.16 | 0.74 | - | - | 1.51±0.37 |
| cfDNA (pg/ml) | 551.1 | 353.8 | - | - | 291.9±97.4 |
| | | | | VTE | |

**Fig 3. Coagulation profile of SCLC patients who developed a VTE during the study period compared to all other patients.** Data is presented as mean ± SD where applicable. Patients 3 and 4 did not donate all blood samples and for P1, thrombin generation during treatment could not be assessed due to treatment with anticoagulants.

hypercoagulable activity, which is also supported by the fact that the activity of both factors is increased and correlated. The changes during the treatment were modest only showing a reduced lagtime for patients with ED, and increased FVIII in patients with LD.

In agreement with our findings, Gezelius et al. found EV TF to be significantly higher in the ED patients when compared to the LD group in a large cohort of SCLC patients [46]. Debaugnies et al [47] and Königsbrügge et al [48] demonstrated that TG in cancer patients was higher than in healthy controls. The majority of patients enrolled in these two studies were diagnosed with malignancies associated with a modest to high risk of VTE [47, 48]. Nielsen et al also found an increased TG and PPL activity in patients with multiple myeloma [44]. Tripodi et al. found that an increase in FVIII and an increase in protein C was associated with a hypercoagulable profile in patients with Cushing disease [18].

We hypothesized that EVs were responsible for an increased coagulation activity in the SCLC patients, but we did not find an increased TG after addition of EVs from the cancer patients to a standard plasma. However, we observed a trend towards an increased PPL activity of EVs, and PPL activity of EVs correlated with TG of the pellets and with PPL in plasma indicating that the main PPL activity is generated of EVs. Although the total number of EVs was higher in cancer patients, the population found in the patients had a higher proportion of smaller EVs (Fig 2D). TF activity was increased in EVs, surprisingly in both 20K and 100K pellets (Fig 2C). This effect was not apparent in the TG analyses probably because the trigger reagents contained TF. According to previous publications, this effect should only be present in 20K pellet [9, 49–51]. Doormaal et al [7] were not able to detect any differences in either PPL or TF activity associated with EVs between cancer patients and healthy controls. This could partly be explained by the inclusion of patients with different cancer diagnoses, which

may obscure the results, as malignancies with low VTE risks may have limited activity of pro-coagulant EVs [10]. Zwicker et al [13] observed a higher concentration of TF positive EVs in cancer patients with VTE compared to the patients without VTE. In a paper by Tesselaar et al [12], EV-associated TF activity was elevated in only few of the included cancer patients, but the majority had levels similar to healthy controls. Contrarily to this, among patients with multiple myeloma EV-associated TF activity has been reported to be 4-fold higher than that of healthy controls [44, 52].

During the study only four patients developed a VTE (Fig 3), and it was not possible to predict these events from the coagulation measurements, which is in accordance with the RAS-TEN study [46]. The RASTEN study did, however, indicate that it was possible to associate TG peak with an increased mortality. It has earlier been found that the activity of TF was significantly associated with occurrence of VTE, and tumor stage, and/or malignancy grade of the cancer [7, 53–55]. In the CATS study, it was found that an increased TG was significantly related to a high risk of developing VTE [56], but a later study by Thaler et al. showed that the activity of TF was associated with mortality and not with the incidence of VTE [8]. The CATS study encompasses patients with different cancer diagnoses.

cfDNA was measured as a surrogate measurement for NETs. This was increased in the cancer patients, especially in patients with ED, but it was not predictive for VTE. Previously, in a study by Demers, it was found that the formation of NETs correlated positively with both tumour progression and other prothrombotic markers [57]. Moreover, Razak found evidence of a prothrombotic capacity of NETs; they observed that NETs entangled cancerous cells allowing for tumour spread and development of a prothrombotic environment [58].

A limitation of this study is the small group of patients especially of those who developed VTE. However, the measurements of coagulation activity were clearly significant, and in the patients with VTE, we were unable to confirm an increased coagulation activity, which is in accordance with a recently published larger study on patients with SCLC [46].

In conclusion, we found that SCLC patients display an overall increased coagulation activity at time of diagnosis and during the treatment, which may contribute to their high risk of VTE. The increased coagulation activity is triggered by augmented PPL and TF activity present in the EVs but probably also other factors in plasma, such as an increased FVIII. However, the coagulation analyses could not predict the patients' risk of VTE.

## Supporting information

**S1 Table. SCLC Baseline biochemistry and blood count data.**
(PDF)

**S1 Fig. Uncropped and original Immunoelectron microscopy (IEM) analysis of extracellular vesicle marker CD9 performed on a pool of isolated vesicles from all donors.** A) 20K pel CD9 positive vesicles. B) Both CD9 positive and negative vesicles isolated for the 100K pel. C) 100K pel CD9 positive vesicle (S2 Fig).
(TIF)

**S2 Fig. Uncropped and original Western blot analysis against CD9 for the 20K and 100K EV pellets from healthy controls (HC) and small cell lung cancer patients (S2 Fig).**
(TIF)

## Acknowledgments

We thank the Department of Clinical Immunology at Aalborg University Hospital in Denmark for kindly allowing us the use of their ultracentrifuge and facilities.

## Author Contributions

**Conceptualization:** Shona Pedersen, Ursula Falkmer, Søren Risom Kristensen.

**Data curation:** Shona Pedersen, Anne Flou Kristensen.

**Formal analysis:** Shona Pedersen, Anne Flou Kristensen, Gunna Christiansen.

**Funding acquisition:** Ursula Falkmer, Søren Risom Kristensen.

**Investigation:** Shona Pedersen, Anne Flou Kristensen.

**Methodology:** Shona Pedersen, Anne Flou Kristensen, Ursula Falkmer, Søren Risom Kristensen.

**Project administration:** Shona Pedersen, Søren Risom Kristensen.

**Resources:** Shona Pedersen.

**Supervision:** Shona Pedersen, Ursula Falkmer, Søren Risom Kristensen.

**Visualization:** Shona Pedersen.

**Writing – original draft:** Shona Pedersen, Anne Flou Kristensen.

**Writing – review & editing:** Shona Pedersen, Anne Flou Kristensen, Ursula Falkmer, Gunna Christiansen, Søren Risom Kristensen.

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
