## [Decision Letter · Decision Letter 0]

11 Mar 2021

PONE-D-21-04828

Increased activity of procoagulant factors in patients with small cell lung cancer

PLOS ONE

Dear Dr. Pedersen,

Thank you for submitting your manuscript to PLOS ONE. After careful consideration, we feel that it has merit but does not fully meet PLOS ONE’s publication criteria as it currently stands. Therefore, we invite you to submit a revised version of the manuscript that addresses the points raised during the review process.

We look forward to receiving your revised manuscript.

Kind regards,

Hugo ten Cate, MD, PhD

Academic Editor

PLOS ONE

Journal Requirements:

2. Thank you for including your ethics statement:  "The study was approved by the regional Ethical Committee (N-20140055) and in accordance with the Declaration of Helsinki. ".   

5. Thank you for stating the following in the 'Financial disclosure statement' Section of your manuscript:

"This work was funded by grants from the Danish Research Council for 512 Independent Research (4183-

513 00268) and the Obel Family Foundation (26145)."

Additional Editor Comments:

The expert reviewers raise a fairly large number of issues that need to be addressed and a major revision of this manuscript is therefore required before further decisions as to potential publication can be made.

Reviewers' comments:

Reviewer's Responses to Questions

**Comments to the Author**

1. Is the manuscript technically sound, and do the data support the conclusions?

Reviewer #1: No

Reviewer #2: Yes

2. Has the statistical analysis been performed appropriately and rigorously? 

Reviewer #1: Yes

Reviewer #2: Yes

3. Have the authors made all data underlying the findings in their manuscript fully available?

Reviewer #1: Yes

Reviewer #2: Yes

4. Is the manuscript presented in an intelligible fashion and written in standard English?

Reviewer #1: Yes

Reviewer #2: Yes

5. Review Comments to the Author

Reviewer #1: 1/ Page 5. “which is considered to be the main initiator of coagulation”. TF is the main initiator of coagulation.

2/ Page 5. Ref 11 is quite old. A more recent review on TF that discusses its role in thrombosis in PMID:29437578.

3/ Page 5. Ref 7,12, 13. PMID:23798713 should be added when discussing TF+ EVs and thrombosis in cancer.

4/ Page 5 Ref 8. Two other studies have reported an association between EV TF activity and survival in cancer patients PMID:23856554; PMID:29539580.

5/ Page 5 FVIII and risk of cancer-associated thrombosis. PMID:31945549 should be added.

6/ Page 5 “Thus, an increase in levels of EVs containing TF and PPL may play a crucial role for the risk of VTEs in cancer patients”. Additional references should be added. PMID:23798713; PMID+28807983.

7/ Page 5. More recent references describing the role of NETs in cancer are PMID:24590420; PMID31315434.

8/ Page 6. There are other sources of cfDNA in plasma. cfDNA is not a good marker of NETs. This section should be re-written.

9/ Page 6. There is some controversy about the capacity of cfDNA to activation coagulation- see PMID:27919911.

10/ Page 6. Circulating cancer cells are very rare and it is unlikely that they play a major role in occlusion of vessels.

11/ Page 6. The term extracellular vesicles is recommended to describe all forms of vesicles. These can be subdivided into large (microvesicles or microparticles) or small (exosomes) vesicles. This terminology should be introduced.

12/ Page 7. The characteristics of the control group should be added to table 1. It is not sufficient to simply describe the median age and range.

13/ Page 7. “first few millilitres were discarded”. Does this mean the first tube?

14/ Page 9. Ref 28 describes an assay to measure EV TF activity in mice. This should be replaced by PMID:30656275.

15/ Page 9. “TF-1” should be “HTF-1”.

16/ Page 12. The type of statistical test used for the analysis of the data should be added to the tables.

17/ Page 14. “platelet-poor plasma” should be changed to “PPP”.

18/ Page 15. It is not clear if the top part of Table 2 shows baseline values for the patients compared with healthy controls or includes all samples? Only baseline values should be used in this comparison.

19/ Page 16. The text states that TF activity was significantly increased in the patients compared with the healthy controls. However, Table 2 does not indicated a significant increase. This appears to be an error in the Table.

20/ Page 17. The data investigating the effect of adding back isolated vesicles to standard plasma should be removed. The reason for isolating vesicles from plasma is to remove them from the numerous inhibitors that are present in plasma. Adding them back to plasma makes no sense.

21/ Page 18. A limitation of the study is the small number of thrombotic events (=4). This should be acknowledged.

22/ Page 20. The Gezelius paper (ref 32) did not have a control group. It did show higher levels of EV TF activity in ED patients compared with LD patients.

23/ Page 20. PMID:21444402 also analyzed TG in cancer patients and should be referenced.

24/ Page 21. Do the authors have any evidence that large EVs are lost?

25/ Page 21. Doormaal is listed as ref 11 but is not present in the reference list.

26/ Page 21. Zwicker et al ref 13 did not include colorectal cancer patients.

27/ Page 21. Contrarily is not a word.

Reviewer #2: General comments: Language is in part quite vague, especially in introduction.

Can you really declare plasma pooled if it's from one patient? Might not serve as an ideal control due to the high variability of coagulation activity between individuals.

Corrections:

104: the activated coagulation factors V and VIII (FVa and FVIIIa, respectively).

109: PPL AND TF activity.

117: patients WERE enrolled

122: OF INR and...

124: technically, ASA and clopidogrel is no anticoagulant therapy

127: Better phrasing: Occurrence of intracranial haemorrhage 3 months prior to start of the study

128: What are effective contraceptives?

132: patient files

181: Is thus the right word here?

287: Table 1. Are T-. N- and M-stage explained somewhere? If not please explain inFigure legend, or leave out if not relevant for the manuscript

376 there is no figure 2E

Wouldn’t it make sense to rearrange fig 2 so it’ll fit the order in the text?

Figure legend format is different from fig 1 at least in the pdf received.

Figure 2a, I get what you’re showing but it’s a little hard to distinguish the traces.

6. PLOS authors have the option to publish the peer review history of their article (what does this mean?). If published, this will include your full peer review and any attached files.

Reviewer #1: No

Reviewer #2: No

---

## [Author Response · Author response to Decision Letter 0]

23 Apr 2021

Dear Editor and Reviewers,

We thank you for the time and effort in reviewing our manuscript. We believe that the advised revisions and comments has heightened the scientific content and coherence of our article. In the document below, we have addressed and responded to the reviewer’s and editors’ comments - point-by-point. Throughout the manuscript, revisions are clarified using track changes mode. 

In accordance to point 4, we have also included, under Supporting Information files, the original and uncropped Immunoelectron microscopy (IEM) images (see S2_raw_IEM images.pdf) and western blots (see S3_raw_WB.pdf).

In accordance to point 5, we have removed the ‘financial disclosure statement’ from the manuscript.

We would like to state under Financial disclosure statement that: 

1. “This work was funded by grants from the Danish Research Council for Independent Research, 4183-00268), https://ufm.dk/ (S.R.K.); and the Obel Family Foundation, 26145, http://www.european-funding-guide.eu/scholarship/7862-obel-family-foundation

(S.R.K.).

2. “The funders had no role in study design, data collection and analysis, decision to

publish, or preparation of the manuscript.”

 On behalf of the authors, we would like to express our appreciation for reviewing our manuscript.

Best regards,

Shona Pedersen, Senior Scientist, Associate Professor

Dep. of Clinical Biochemistry and Clinical Medicine 

Aalborg University hospital, Aalborg University

Aalborg, Denmark

PONE-D-21-04828

Increased activity of procoagulant factors in patients with small cell lung cancer

PLOS ONE

Dear Dr. Pedersen,

Thank you for submitting your manuscript to PLOS ONE. After careful consideration, we feel that it has merit but does not fully meet PLOS ONE’s publication criteria as it currently stands. Therefore, we invite you to submit a revised version of the manuscript that addresses the points raised during the review process.

Thank you for the opportunity to submit a revised paper. 

We look forward to receiving your revised manuscript.

Kind regards,

Hugo ten Cate, MD, PhD

Academic Editor

PLOS ONE

Journal Requirements:

2. Thank you for including your ethics statement: "The study was approved by the regional Ethical Committee (N-20140055) and in accordance with the Declaration of Helsinki. ". 

Please amend your current ethics statement to include the full name of the ethics committee/institutional review board(s) that approved your specific study. Thank for pointing this out. Our ethics statement has been revised to “Regional ethics committee for Northern Jutland” and this statement has also been included at the beginning of the Methods section.

We apologize for this fault. We have included the missing data under Supporting Information files, S1 Table. SCLC Baseline biochemistry and blood count data.

5. Thank you for stating the following in the 'Financial disclosure statement' Section of your manuscript:

"This work was funded by grants from the Danish Research Council for 512 Independent Research (4183-

513 00268) and the Obel Family Foundation (26145)."

Please remove any funding-related text from the manuscript and let us know how you would like to update your Funding Statement. Currently, your Funding Statement reads as follows: We apologize for this error. We have deleted the funding-related text from the manuscript.

Additional Editor Comments:

The expert reviewers raise a fairly large number of issues that need to be addressed and a major revision of this manuscript is therefore required before further decisions as to potential publication can be made.

Reviewers' comments:

Reviewer's Responses to Questions

Comments to the Author

1. Is the manuscript technically sound, and do the data support the conclusions?

Reviewer #1: No

Reviewer #2: Yes

2. Has the statistical analysis been performed appropriately and rigorously? 

Reviewer #1: Yes

Reviewer #2: Yes

3. Have the authors made all data underlying the findings in their manuscript fully available?

Reviewer #1: Yes

Reviewer #2: Yes

4. Is the manuscript presented in an intelligible fashion and written in standard English?

Reviewer #1: Yes

Reviewer #2: Yes

5. Review Comments to the Author

We thank the reviewers for their time reviewing this paper and for their many insightful comments. We have revised the paper according to theses comments. Below are our responses to the specific points:

Reviewer #1: 

1/ Page 5. “which is considered to be the main initiator of coagulation”. TF is the main initiator of coagulation. We agree, and this has now been revised in page 5

2/ Page 5. Ref 11 is quite old. A more recent review on TF that discusses its role in thrombosis in PMID:29437578. Grover SP, Mackman N. Tissue Factor: An Essential Mediator of Hemostasis and Trigger of Thrombosis. Arterioscler Thromb Vasc Biol. United States; 2018;38(4):709–25-PMID:29437578, has now been replaced as ref 11 (page 5). 

3/ Page 5. Ref 7,12, 13. PMID:23798713 should be added when discussing TF+ EVs and thrombosis in cancer. Geddings JE, Mackman N. Tumor-derived tissue factor – positive microparticles and venous thrombosis in cancer patients. Blood. 2013;122(11):1873–80-PMID: 23798713, has now been added as ref 14 (page 5).

4/ Page 5. Ref 8. Two other studies have reported an association between EV TF activity and survival in cancer patients PMID:23856554; PMID:29539580. Bharthuar A, Khorana A a, Hutson A, Wang J-G, Key NS, Mackman N, et al. Circulating microparticle tissue factor, thromboembolism and survival in pancreaticobiliary cancers. Thromb Res. Elsevier B.V.; 2013;132(2):180–4- PMID:23856554 and Hisada Y, Thålin C, Lundström S, Wallén H, Mackman N. Comparison of microvesicle tissue factor activity in non-cancer severely ill patients and cancer patients. Thromb Res. 2018;165:1–5- PMID: 29539580, has now been included as reference 15 and 16, respectively (page 5).

5/ Page 5 FVIII and risk of cancer-associated thrombosis. PMID:31945589 should be added. Moik F, Posch F, Grilz E, Scheithauer W, Pabinger I, Prager G, et al. Haemostatic biomarkers for prognosis and prediction of therapy response in patients with metastatic colorectal cancer. Thromb Res. 2020;187:9–17. ref 22 (page 5).

6/ Page 5 “Thus, an increase in levels of EVs containing TF and PPL may play a crucial role for the risk of VTEs in cancer patients”. Additional references should be added. PMID:23798713; PMID:28807983.

Hisada Y, Mackman N. Cancer-associated pathways and biomarkers of venous thrombosis. Blood. 2017;130(13):1499–506. -PMID: 23798713 and PMID:28807983, has now been added as ref 25 and 14 (page 5).

7/ Page 5. More recent references describing the role of NETs in cancer are PMID:24590420; PMID:31315434. Demers M, Wagner DD. NETosis: a new factor in tumor progression and cancer-associated thrombosis. Semin Thromb Hemost. 2014;40(3):277–83- PMID: 24590420 and Thålin C, Hisada Y, Lundström S, Mackman N, Wallén H. Neutrophil Extracellular Traps: Villains and Targets in Arterial, Venous, and Cancer-Associated Thrombosis. Arterioscler Thromb Vasc Biol. 2019;39(9):1724–38- PMID: 31315434, has now been added as ref 27 and 28 (page 5).

8/ Page 6. There are other sources of cfDNA in plasma. cfDNA is not a good marker of NETs. This section should be re-written. We agree with the reviewer that cfDNA is not a very good marker. On the other hand we have no perfect markers, and cfDNA has been used as a marker in many other studies (as e.g. described in Thålin et al, ref 28). To-day we would perhaps have made another choice but this was what was decided to measure some years ago when the project was planned. We have rewritten the section. 

9/ Page 6. There is some controversy about the capacity of cfDNA to activation coagulation- see PMID:27919911. Please see the former point.

10/ Page 6. Circulating cancer cells are very rare and it is unlikely that they play a major role in occlusion of vessels. We agree, and we have removed the sentence.

11/ Page 6. The term extracellular vesicles is recommended to describe all forms of vesicles. These can be subdivided into large (microvesicles or microparticles) or small (exosomes) vesicles. This terminology should be introduced. The term EVs, including microvesicles and exosomes, has now been introduced and described in page 6. Important references to support the EV description has also been included (ref 35-36) page 6.

12/ Page 7. The characteristics of the control group should be added to table 1. It is not sufficient to simply describe the median age and range. We have now added this information to Table 1. 

13/ Page 7. “first few millilitres were discarded”. Does this mean the first tube? Yes. This has now been corrected (page 7) 

14/ Page 9. Ref 28 describes an assay to measure EV TF activity in mice. This should be replaced by PMID:30656275. Hisada Y, Mackman N. Measurement of tissue factor activity in extracellular vesicles from human plasma samples. Res Pract Thromb Haemost. 2019;3(1):44–8- PMID:30656275, has now been added as ref 41 (page 9).

15/ Page 9. “TF-1” should be “HTF-1”. This has now been revised on page page 9.

16/ Page 12. The type of statistical test used for the analysis of the data should be added to the tables. The statistic applied are now included in the Table text for Table 2 and 3.

17/ Page 14. “platelet-poor plasma” should be changed to “PPP”. This has now been revised in page 14.

18/ Page 15. It is not clear if the top part of Table 2 shows baseline values for the patients compared with healthy controls or includes all samples? Only baseline values should be used in this comparison. The table includes initial data on controls versus SCLC baseline patients. Moreover, the SCLC patients were further divided into limited (LD) or extended (ED) disease and followed during treatment and follow-up. This has now been clarified in the table. An additional table 2 text now included to clarify this (page 15).

19/ Page 16. The text states that TF activity was significantly increased in the patients compared with the healthy controls. However, Table 2 does not indicated a significant increase. This appears to be an error in the Table. Yes, it was correct that there was 9-fold increase in TF activity, but this was only borderline significant (p=0.51), due to the variation. This has now been explained on page 16. 

20/ Page 17. The data investigating the effect of adding back isolated vesicles to standard plasma should be removed. The reason for isolating vesicles from plasma is to remove them from the numerous inhibitors that are present in plasma. Adding them back to plasma makes no sense. We agree with the reviewer that they do not describe very exciting results, and especially the results in fig 2A were disappointing. However, we do not agree that it makes no sense to add them to a standard plasma. In this environment, we have “normalized” everything except EVs, and therefore we can compare the effect of these. We have done this before in another population, patients with malignant myeloma (Plos One: doi.org/10.1371/journal.pone.0210835), where we saw a substantial effect. We agree and admit that there is some overlap with the other measurements (TF and PPL), but we think that the results supplement each other. We were surprised that we did not find a difference in TG, because we had found the opposite in the previous paper on malignant myeloma. Since these results also represent a considerable amount of work, we would prefer to keep this figure in the paper. However, if the editor prefers to discard the figure, we are of course willing to do so.

21/ Page 18. A limitation of the study is the small number of thrombotic events (=4). This should be acknowledged. We certainly agree, but this was initially mentioned in the paper (page 22).

22/ Page 20. The Gezelius paper did not have a control group. It did show higher levels of EV TF activity in ED patients compared with LD patients. Yes, it is correct that the Gezelius et al. paper did not include control persons in their study. We apologize for the mistake, but our findings on significantly increased EV TF activity in the LD patients, is comparable. This has been changed in the paper.

23/ Page 20. PMID:21444402 also analyzed TG in cancer patients and should be referenced. We are sorry, but It was not possible to locate the article with the advised PMID number.

24/ Page 21. Do the authors have any evidence that large EVs are lost? No, it was pure speculation. It has now been removed from the article.

25/ Page 21. Doormaal is listed as ref 11 but is not present in the reference list. Thank you for pointing this out. Doormaal is now included as reference 7, this has now been rectified in the main text.

26/ Page 21. Zwicker et al ref 13 did not include colorectal cancer patients. Thank you for pointing this out. Cancer patients of different histologies, with and without evidence of acute VTE (venous thromboembolism), were included in this study. This has now been corrected to “Zwicker et al [13] observed a higher concentration of TF positive EVs in cancer patients with VTE compared to the patients without VTE”

27/ Page 21. Contrarily is not a word. According to the Cambridge English dictionary this is an English word meaning “in a way opposite of something”

Reviewer #2: General comments: Language is in part quite vague, especially in introduction.

Can you really declare plasma pooled if it's from one patient? Might not serve as an ideal control due to the high variability of coagulation activity between individuals. Thank you for your comments and suggestions. We have tried to strengthen the language and made several, although smaller changes. We agree with the reviewer’s point regarding “pooled plasma”. It was not a pooled plasma when it originated from just one person, and we have changed it to “Standard plasma”. It can be argued that a real normal plasma from more persons would be preferable, but it is only used as a basis for comparison of EVs from the different patients, and, therefore, we think that it is OK to use this standard plasma as we call it.

Corrections:

104: the activated coagulation factors V and VIII (FVa and FVIIIa, respectively). This has now been corrected (page 6).

109: PPL AND TF activity. This has now been corrected (page 6).

117: patients WERE enrolled This has now been corrected (page 7).

122: OF INR and... This has now been corrected (page 7).

124: technically, ASA and clopidogrel is no anticoagulant therapy. Yes, we agree. This has been corrected to “(platelet inhibitors, ASA and clopidogrel were allowed),” in page 7.

127: Better phrasing: Occurrence of intracranial haemorrhage 3 months prior to start of the study. Thank you, this has now been rephrased, accordingly (page 7).

128: What are effective contraceptives? Thank you for pointing this out, “effective” has been removed from the sentence, and it has been changed to oral contraceptives (page 7).

132: patient files This has now been corrected (page 7).

181: Is thus the right word here? “Thus” has been removed from the sentence (page 9). 

287: Table 1. Are T-. N- and M-stage explained somewhere? If not please explain in Figure legend, or leave out if not relevant for the manuscript. Yes, TMN (Tumor, lymph Node, and Metastasis) staging has now been mentioned (and defined) in the result section and in the table text for Table 1.

376 there is no figure 2E We apologize for this fault - this has now been corrected in the figure text ( line 402)

Wouldn’t it make sense to rearrange fig 2 so it’ll fit the order in the text? Yes, we agree. We have changed the text to match the corresponding figures.

Figure legend format is different from fig 1 at least in the pdf received. The legend style for figures 1 and 2 is the same (at least in our Word-paper).

Figure 2a, I get what you’re showing but it’s a little hard to distinguish the traces. Yes, we agree that the various curves are difficult to distinguish because they are not very different. However, these were the attained results, and the impression from the figure is exactly that they are; hardly different, so we hope that it is OK.

6. PLOS authors have the option to publish the peer review history of their article (what does this mean?). If published, this will include your full peer review and any attached files.

Do you want your identity to be public for this peer review? For information about this choice, including consent withdrawal, please see our Privacy Policy.

Reviewer #1: No

Reviewer #2: No

---

## [Decision Letter · Decision Letter 1]

19 May 2021

PONE-D-21-04828R1

Increased activity of procoagulant factors in patients with small cell lung cancer

PLOS ONE

Dear Dr. Pedersen,

Thank you for submitting your manuscript to PLOS ONE. After careful consideration, we feel that it has merit but does not fully meet PLOS ONE’s publication criteria as it currently stands. Therefore, we invite you to submit a revised version of the manuscript that addresses the points raised during the review process.

We look forward to receiving your revised manuscript.

Kind regards,

Hugo ten Cate, MD, PhD

Academic Editor

PLOS ONE

Journal Requirements:

Additional Editor Comments (if provided):

One reviewer still has some relevant suggestions for improvement of the manuscript, the other reviewer abstains from additional comments.

Reviewers' comments:

Reviewer's Responses to Questions

**Comments to the Author**

1. If the authors have adequately addressed your comments raised in a previous round of review and you feel that this manuscript is now acceptable for publication, you may indicate that here to bypass the “Comments to the Author” section, enter your conflict of interest statement in the “Confidential to Editor” section, and submit your "Accept" recommendation.

Reviewer #2: (No Response)

2. Is the manuscript technically sound, and do the data support the conclusions?

Reviewer #2: Yes

3. Has the statistical analysis been performed appropriately and rigorously? 

Reviewer #2: Yes

4. Have the authors made all data underlying the findings in their manuscript fully available?

Reviewer #2: Yes

5. Is the manuscript presented in an intelligible fashion and written in standard English?

Reviewer #2: Yes

6. Review Comments to the Author

Reviewer #2: General comments.

- Replace Standard pooled Plasma (SPP) with SP in figures and methods (‘Isolation of extracellular vesicles’),

- The NET paragraph in the introduction contains very bulky language which should be improved, see remarks below.

P5 lines 97-98 Better: Neutrophil extracellular traps (NETs) have been linked to the formation of VTEs and may also play a role in cancer-associated thrombosis.

P6 lines 101-106. . Better something like: Although there is no single marker to determine NETs in plasma, surrogate markers for NETs include plasma levels of cfDNA, citrullinated histone H3 and myeloperoxidase. In addition, cfDNA was described to carry a procoagulant activity on its own and therefore might give insight about NET-associated procoagulant activity in patient plasma samples.

P6 l. 109 has been = have been

P.6 l. 113- 116. Better: A low protC activity was associated with increased mortality in different malignancies, e.g. in non-metastasizing lung cancer (38).

P7 l.127: remove the second 'patients'.

P7 l. 131: age over 18 YEARS

P7 l 140: Pregnant and/ or breast-feeding women and women who did not use oral contraceptives were excluded.

P.18/19 Result section Fig 2. Why do you include a curve for Standard Plasma if you don’t mention it in the text? Maybe explain the relevance of the curve not only in the figure legend. Also replace SPP with SP in the Figure and legends.

7. PLOS authors have the option to publish the peer review history of their article (what does this mean?). If published, this will include your full peer review and any attached files.

Reviewer #2: No

---

## [Author Response · Author response to Decision Letter 1]

26 May 2021

PONE-D-21-04828

Increased activity of procoagulant factors in patients with small cell lung cancer

PLOS ONE

Dear Dr. Pedersen,

Thank you for submitting your manuscript to PLOS ONE. After careful consideration, we feel that it has merit but does not fully meet PLOS ONE’s publication criteria as it currently stands. Therefore, we invite you to submit a revised version of the manuscript that addresses the points raised during the review process.

•A rebuttal letter that responds to each point raised by the academic editor and reviewer(s). You should upload this letter as a separate file labeled 'Response to Reviewers'.

•A marked-up copy of your manuscript that highlights changes made to the original version. You should upload this as a separate file labeled 'Revised Manuscript with Track Changes'.

•An unmarked version of your revised paper without tracked changes. You should upload this as a separate file labeled 'Manuscript'.

We look forward to receiving your revised manuscript.

Kind regards,

Hugo ten Cate, MD, PhD

Academic Editor

PLOS ONE

Journal Requirements:

Additional Editor Comments (if provided):

One reviewer still has some relevant suggestions for improvement of the manuscript, the other reviewer abstains from additional comments.

Reviewers' comments:

Reviewer's Responses to Questions

Comments to the Author

1. If the authors have adequately addressed your comments raised in a previous round of review and you feel that this manuscript is now acceptable for publication, you may indicate that here to bypass the “Comments to the Author” section, enter your conflict of interest statement in the “Confidential to Editor” section, and submit your "Accept" recommendation.

Reviewer #2: (No Response)

2. Is the manuscript technically sound, and do the data support the conclusions?

Reviewer #2: Yes

3. Has the statistical analysis been performed appropriately and rigorously? 

 Reviewer #2: Yes

 4. Have the authors made all data underlying the findings in their manuscript fully available?

The PLOS Data policy requires authors to make all data underlying the findings described in their manuscript fully available without restriction, with rare exception (please refer to the Data Availability Statement in the manuscript PDF file). The data should be provided as part of the manuscript or its supporting information or deposited to a public repository. For example, in addition to summary statistics, the data points behind means, medians and variance measures should be available. If there are restrictions on publicly sharing data—e.g. participant privacy or use of data from a third party—those must be specified.

 Reviewer #2: Yes

 5. Is the manuscript presented in an intelligible fashion and written in standard English?

 Reviewer #2: Yes

6. Review Comments to the Author

 Reviewer #2: General comments.

- Replace Standard pooled Plasma (SPP) with SP in figures and methods (‘Isolation of extracellular vesicles’),

Thank you for pointing this out. SPP has now been changed to SP in the actual Figure 2, the associated Figure legend and in the results describing SP (page 17, Line 377).

- The NET paragraph in the introduction contains very bulky language which should be improved, see remarks below. 

P5 lines 97-98 Better: Neutrophil extracellular traps (NETs) have been linked to the formation of VTEs and may also play a role in cancer-associated thrombosis. 

Thank you for rephrasing this sentence. This has now been added to the final manuscript, page 5, line 98. 

P6 lines 101-106. . Better something like: Although there is no single marker to determine NETs in plasma, surrogate markers for NETs include plasma levels of cfDNA, citrullinated histone H3 and myeloperoxidase. In addition, cfDNA was described to carry a procoagulant activity on its own and therefore might give insight about NET-associated procoagulant activity in patient plasma samples.

Thank you for a better phrasing on the information on NETs. This has now been added to the final manuscript, page 5, line 100-104.

P6 l. 109 has been = have been 

This has now been changed.

P.6 l. 113- 116. Better: A low protC activity was associated with increased mortality in different malignancies, e.g. in non-metastasizing lung cancer (38).

Thank you for rephrasing this sentence. This has now been added to the final manuscript, page 5. 

P7 l.127: remove the second 'patients'. 

This has now been removed.

P7 l. 131: age over 18 YEARS 

This has now been changed from age ≥ 18 to age over 18 years.

P7 l 140: Pregnant and/ or breast-feeding women and women who did not use oral contraceptives were excluded.

Thank you for rephrasing this sentence. This has now been changed in the final manuscript.

P.18/19 Result section Fig 2. Why do you include a curve for Standard Plasma if you don’t mention it in the text? Maybe explain the relevance of the curve not only in the figure legend. Also replace SPP with SP in the Figure and legends.

SPP has been replaced by SP in the figure and the legends (page 18). The thrombograms shown in Fig 2A is included to demonstrate that SP in the absence of EVs indicates longer lagtime with a reduced peak height, whereas controls and SCLC patient samples spiked with EVs, reflected shorter lagtime with higher peak height. We have also included additional text (page 17)

7. PLOS authors have the option to publish the peer review history of their article (what does this mean?). If published, this will include your full peer review and any attached files.

Do you want your identity to be public for this peer review? For information about this choice, including consent withdrawal, please see our Privacy Policy.

Reviewer #2: No

---

## [Decision Letter · Decision Letter 2]

9 Jun 2021

Increased activity of procoagulant factors in patients with small cell lung cancer

PONE-D-21-04828R2

Dear Dr. Pedersen,

We’re pleased to inform you that your manuscript has been judged scientifically suitable for publication and will be formally accepted for publication once it meets all outstanding technical requirements.

Kind regards,

Christophe Leroyer

Academic Editor

PLOS ONE

**Comments to the Author**

1. If the authors have adequately addressed your comments raised in a previous round of review and you feel that this manuscript is now acceptable for publication, you may indicate that here to bypass the “Comments to the Author” section, enter your conflict of interest statement in the “Confidential to Editor” section, and submit your "Accept" recommendation.

Reviewer #2: All comments have been addressed

2. Is the manuscript technically sound, and do the data support the conclusions?

Reviewer #2: (No Response)

3. Has the statistical analysis been performed appropriately and rigorously? 

Reviewer #2: (No Response)

4. Have the authors made all data underlying the findings in their manuscript fully available?

Reviewer #2: (No Response)

5. Is the manuscript presented in an intelligible fashion and written in standard English?

Reviewer #2: (No Response)

6. Review Comments to the Author

Reviewer #2: (No Response)

7. PLOS authors have the option to publish the peer review history of their article (what does this mean?). If published, this will include your full peer review and any attached files.

Reviewer #2: No

---

## [Editor Report · Acceptance letter]

7 Jul 2021

PONE-D-21-04828R2 

Increased activity of procoagulant factors in patients with small cell lung cancer 

Dear Dr. Pedersen:

I'm pleased to inform you that your manuscript has been deemed suitable for publication in PLOS ONE. Congratulations! Your manuscript is now with our production department. 

Kind regards, 

on behalf of

Dr. Christophe Leroyer 

Academic Editor

PLOS ONE